# The Ki-67 and RepoMan mitotic phosphatases assemble via an identical, yet novel mechanism

Ganesan Senthil Kumar[1], Ezgi Gokhan[2], Sofie De Munter[3], Mathieu Bollen[3], Paola Vagnarelli[2], Wolfgang Peti[1], Rebecca Page[4]*

[1]Department of Molecular Pharmacology, Physiology and Biotechnology, Brown University, Providence, United States; [2]College of Health and Life Science, Research Institute for Environment, Health and Society, Brunel University London, Uxbridge, United Kingdom; [3]Laboratory of Biosignaling and Therapeutics, Department of Cellular and Molecular Medicine, KU Leuven, Leuven, Belgium; [4]Department of Molecular Biology, Cell Biology and Biochemistry, Brown University, Providence, United States

**Abstract** Ki-67 and RepoMan have key roles during mitotic exit. Previously, we showed that Ki-67 organizes the mitotic chromosome periphery and recruits protein phosphatase 1 (PP1) to chromatin at anaphase onset, in a similar manner as RepoMan (*Booth et al., 2014*). Here we show how Ki-67 and RepoMan form mitotic exit phosphatases by recruiting PP1, how they distinguish between distinct PP1 isoforms and how the assembly of these two holoenzymes are dynamically regulated by Aurora B kinase during mitosis. Unexpectedly, our data also reveal that Ki-67 and RepoMan bind PP1 using an identical, yet novel mechanism, interacting with a PP1 pocket that is engaged only by these two PP1 regulators. These findings not only show how two distinct mitotic exit phosphatases are recruited to their substrates, but also provide immediate opportunities for the design of novel cancer therapeutics that selectively target the Ki-67:PP1 and RepoMan:PP1 holoenzymes.

*For correspondence:
Rebecca_Page@brown.edu

**Competing interests:** The authors declare that no competing interests exist.

## Introduction

Mitotic exit comprises a complex series of events that includes sister chromatid segregation, mitotic spindle disassembly, nuclear-envelope re-assembly and chromosome decondensation (*Wurzenberger and Gerlich, 2011*). How these events are coordinated during mitotic exit is still an open question. The emerging picture is that exiting mitosis requires the specific engagement and activation of protein phosphatases, including ser/thr phosphatase protein phosphatase 1 (PP1) (*Funabiki and Wynne, 2013*; *Rosenberg et al., 2011*). While PP1 exhibits broad specificity, it acts in a highly specific manner by forming stable complexes, known as holoenzymes, with a host of regulatory proteins that direct PP1 activity towards specific substrates and localize PP1 to specific regions of the cell (*Hendrickx et al., 2009*; *Bollen et al., 2010*; *Peti et al., 2013*; *Peti and Page, 2015*; *Choy et al., 2014*; *O'Connell et al., 2012*). A detailed understanding of how key PP1 regulators bind and direct PP1 activity during distinct stages of the cell cycle is still largely missing.

Ki-67 is widely used as a prognostic marker for many cancers (*Dowsett et al., 2011*; *Lin et al., 2016*; *Sobecki et al., 2016*), yet for decades, its molecular function remained largely unknown. Recently, it was shown that one role of Ki-67 is to function as a 'DNA surfactant', preventing individual chromosomes from collapsing into a single chromatin mass upon nuclear envelope disassembly by binding directly to the surface of chromatin (*Cuylen et al., 2016*). A second recently discovered

function is that it binds and regulates the activity of PP1 during mitosis using the canonical PP1 RVxF small linear motif (SLiM) (*Booth et al., 2014*). The only other protein that exhibits any sequence similarity with Ki-67 near its RVxF motif is RepoMan (recruits PP1γ onto mitotic chromatin at anaphase, also known as cell division cycle associated 2, CDCA2), a nuclear-specific protein that was discovered for its ability to specifically recruit PP1γ to chromosomes at anaphase onset and, like Ki-67, is upregulated in many cancers (*Trinkle-Mulcahy et al., 2006*; *Vagnarelli, 2014*). During mitotic exit, PP1, in particular PP1γ (PP1 isoforms include PP1α, PP1β, PP1γ and PP1γ2; ≥ 85% identity between isoforms), is essential for histone dephosphorylation, nuclear-envelope reassembly and chromatin remodeling (*Qian et al., 2011*; *Vagnarelli et al., 2006*; *Wurzenberger et al., 2012*; *Qian et al., 2013*; *Vagnarelli et al., 2011*). These PP1γ-specific processes are mediated largely by the Ki-67: PP1γ and RepoMan:PP1γ holoenzymes. Two key unresolved questions are how do these regulators assemble mitotic phosphatases and how are these interactions dynamically regulated in both space and time? Answers to these questions will provide novel opportunities for the development of Ki-67: PP1 and RepoMan:PP1 specific therapeutics for cancer. In this report, we demonstrate how Ki-67 and RepoMan form mitotic exit phosphatases, how they distinguish between distinct PP1 isoforms and how the assembly of these two holoenzymes is dynamically regulated by Aurora B kinase during mitosis.

## Results

### The Ki-67 and RepoMan PP1 interaction domain

Ki-67 (3256 aa) and RepoMan (1023 aa) bind PP1γ to form isoform-specific holoenzymes (*Booth et al., 2014*; *Trinkle-Mulcahy et al., 2006*). Previously, we showed that these proteins exhibit sequence similarity in only a very short region, of about 40 residues (*Figure 1A*) (*Booth et al., 2014*). This region includes the canonical RVxF SLiM that is critical for PP1 binding ($^{505}$RVSF$^{508}$, Ki-67; $^{392}$RVTF$^{395}$, RepoMan; more than 70% of PP1 regulators contain the RVxF SLiM) (*Trinkle-Mulcahy et al., 2006*; *Vagnarelli et al., 2011*). Because recent efforts to understand how PP1 activity is directed by its >200 distinct regulators has revealed that residues outside the RVxF motif are also essential for PP1 holoenzyme formation and function (*Peti et al., 2013*; *O'Connell et al., 2012*; *Terrak et al., 2004*), we reasoned that this entire region was critical for PP1 binding. Using isothermal titration calorimetry (ITC), we showed that Ki-67$_{496-536}$ and RepoMan$_{383-423}$ bind PP1γ$_{7-323}$ with a 1:1 stoichiometry and with nearly equivalent affinities (*Figure 1B*, left; *Figure 1—figure supplement 1*; $K_D$ = 193 ± 16 nM and 133 ± 5 nM; respectively; all ITC experiments are summarized in *Table 1*). We also showed that extending this domain does not enhance binding (*Table 1*). Finally, NMR spectroscopy experiments with RepoMan confirmed that all residues in this conserved region interact with PP1 (*Figure 1—figure supplements 2,3*). Together, these data suggest that Ki-67$_{496-536}$ and RepoMan$_{383-423}$ bind PP1 using identical mechanisms and that this conserved region, which extends beyond the RVxF SLiM, constitutes the full PP1 interaction domain.

### The discovery of a novel PP1 interaction SLiM, the KiR-SLiM

The Ki-67/RepoMan residues C-terminal to the canonical RVxF motif are not present in any other PP1 regulator whose holoenzyme structure is known, suggesting these regulators bind PP1 using a novel mechanism. To identify this mechanism, we determined the crystal structures of the Ki-67$_{496-536}$:PP1γ$_{7-308}$ and RepoMan$_{383-423}$:PP1γ$_{7-308}$ holoenzymes complexes to 2.0 Å and 1.3 Å, respectively (*Figure 1C,D*; *Figure 1—source data 1*). As predicted by the ITC and NMR studies, Ki-67 and RepoMan bind PP1 using an identical mechanism. Namely, Ki-67 and RepoMan form a classical β-hairpin on the top of PP1 that extends from the PP1 RVxF binding pocket towards the PP1 N-terminus and then back again (*Figure 1D*; ~2600 Å of buried surface area). The structures also show that two established PP1-specific SLiMs—the RVxF-SLiM (*Egloff et al., 1997*), $^{505}$RVSF$^{508}$$_{Ki-67}$/$^{392}$RVTF$^{395}$$_{RM}$ and the ΦΦ-SLiM (*O'Connell et al., 2012*), $^{515}$EL$^{516}$$_{Ki-67}$/$^{402}$EV$^{403}$$_{RM}$ —bind directly to PP1 in the RVxF and ΦΦ binding pockets (*Figure 1D*). However, as predicted, the Ki-67/RepoMan residues C-terminal to these SLiMs (aa 517–535/404–422) also bind PP1 and do so in a manner never observed for any other PP1 regulator (*Figure 1D,E*). Residues F517$_{Ki-67}$/F404$_{RM}$ and P523$_{Ki-67}$/A410$_{RM}$ are anchored to PP1γ via two hydrophobic pockets while multiple residues in Ki-67/RepoMan bind the side chains of R74$_{PP1}$, Y78$_{PP1}$ and Q294$_{PP1}$ via polar and salt bridge interactions. This

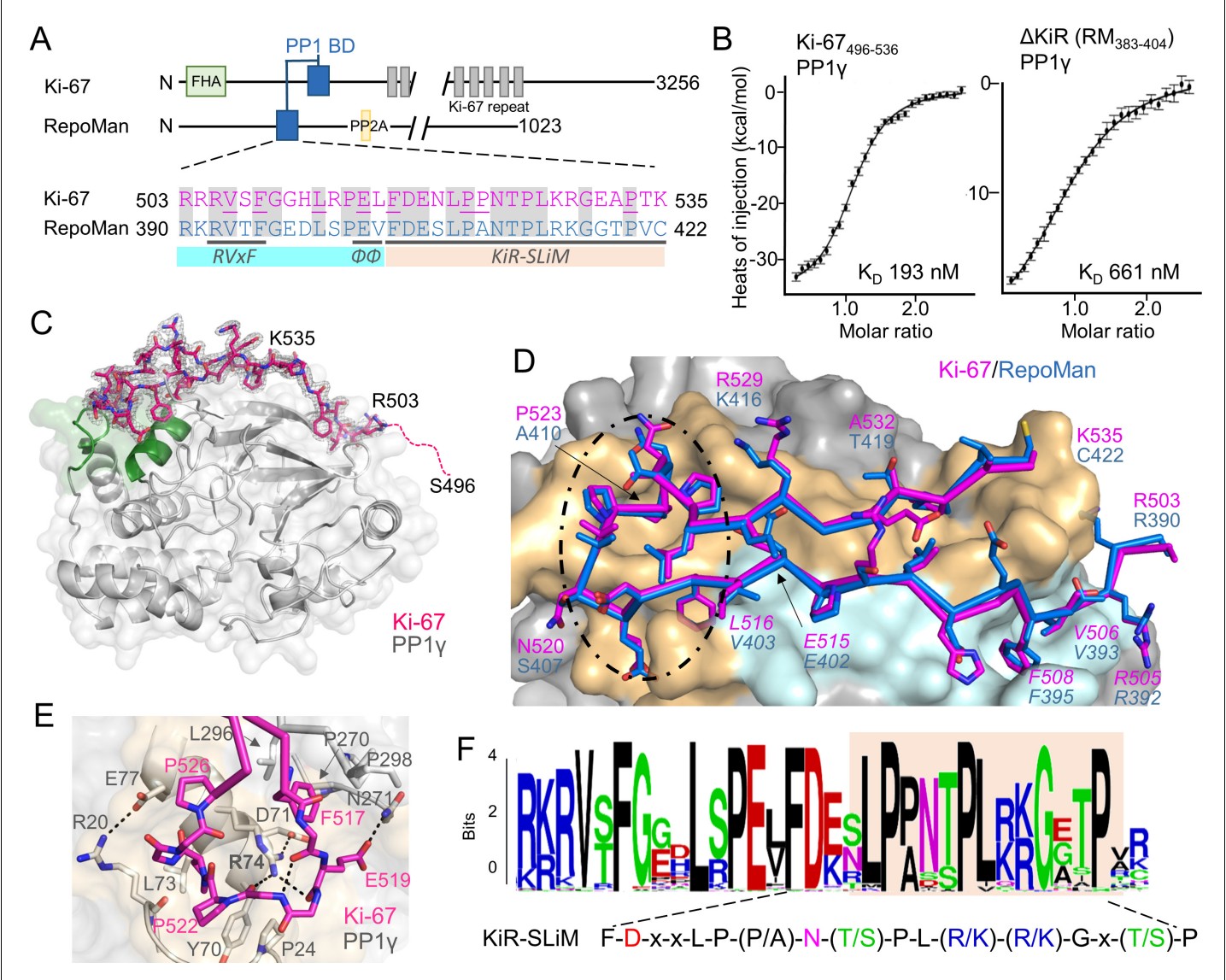

**Figure 1.** The RepoMan:PP1 holoenzyme complex. (**A**) Cartoon depicting Ki-67 and RepoMan domains. The only region of homology between the two proteins is indicated in blue. The sequences corresponding to the homologous regions are shown below, with conserved residues highlighted in grey. Ki-67 residues that interact directly with PP1 are underlined. The sequences corresponding to the RVxF and ΦΦ SLiMs (blue highlight) and the newly discovered KiR-SLiM (orange) are shown. (**B**) *left*, binding isotherm of Ki-67$_{496-536}$ with PP1γ$_{7-323}$ ($K_D$, 193 ± 16 nM; the $K_D$ of the corresponding domain of RepoMan$_{383-423}$ with PP1γ$_{7-323}$ is 133 ± 16 nM); *right*, binding isotherm of the ΔKiR-SLIM, RepoMan$_{383-404}$, with PP1γ$_{7-323}$ ($K_D$, 661 ± 160 nM). (**C**) Crystal structure of the Ki-67:PP1γ holoenzyme. PP1γ is in grey and Ki-67$_{496-536}$ is in pink with the 2$F_o$–$F_c$ electron density map contoured at 1σ (2.0 Å); no electron density was observed for Ki-67 residues 496–503 (pink dotted line) and 536. PP1 residues in green correspond to PP1 secondary structure elements helix A', loop 1 and helix B. (**D**) Close-up of the Ki-67 (pink) and RepoMan (blue) interaction with PP1. Ki-67 residues 503–516 and RepoMan residues 390–403 bind the PP1 RVxF and ΦΦ binding pockets (cyan surface; the Ki-67 and RepoMan RVxF and ΦΦ SLiM residues are labeled and in italics). Ki-67 residues 517–535 and RepoMan residues 404–422 bind the newly defined KiR-SLiM binding pocket (beige surface). The black dotted line highlights the area shown in **E**. (**E**) The hydrophobic and polar interactions between Ki-67 (pink sticks) and PP1γ (grey sticks; surface). Hydrogen bonds and salt bridge interactions are indicated by dotted lines with the interacting residues labeled. (**F**) HMMER-derived sequence logo of the Ki-67/ RepoMan PP1 binding domain, with the KiR-SLIM highlighted in beige (hydrophobic residues, black; acidic residues, red; basic residues blue; glycine/ serine/threonine, green; asparagine/glutamine, pink).

The following source data and figure supplements are available for figure 1:

**Source data 1.** Data collection and refinement statistics.

**Figure supplement 1.** Isothermal titration calorimetry of Ki-67 and RepoMan with PP1.

*Figure 1 continued on next page*

*Figure 1 continued*

**Figure supplement 2.** The PP1 binding domain of RepoMan is an intrinsically disordered protein (IDP).

**Figure supplement 3.** Identification of the RepoMan minimal PP1 binding domain.

**Figure supplement 4.** RepoMan F404A variant (RepoMan^FA) behaves like wt RepoMan (RepoMan^wt) in cells.

orders PP1γ L1 (*Figure 1C*), a loop that is generally more dynamic in most free PP1 and PP1 holoenzyme structures as evidenced by its higher B-factors and less well-defined electron density. Although $F517_{Ki-67}/F404_{RM}$ are the most buried residues in both complexes, mutating this residue to an alanine does not negatively affect RepoMan function (*Figure 1—figure supplement 4*).

Because these residues are critical for binding (removing them decreases the affinity ~five-fold; *Figure 1B*, right), we termed this novel PP1 interaction SLiM the *KiR-SLiM* (Ki-67-RepoMan SLiM). The general KiR-SLiM motif, FDxxLP(P/A)N(T/S)PL(R/K)(R/K)Gx(T/S)P was determined using HMMER (*Figure 1F*) (*Finn et al., 2015*). Critically, a subsequent search of the UniProtK/Swiss-Prot database (*de Castro et al., 2006*) using the most degenerate version of the KiR-SLiM identified only Ki-67 and RepoMan proteins. This demonstrates that, unexpectedly, only these two PP1 regulators likely use the KiR-SLiM interaction surface on PP1.

**Table 1.** Isothermal titration calorimetry (ITC) measurements.

| Titrant | PP1 | $K_D$ (nM) | repeats |
|---|---|---|---|
| **Ki-67$_{496–536}$** | | | |
| wt | $\alpha_{7-330}$ | 779 ± 142 | 3 |
| wt | $\gamma_{7-323}$ | 193 ± 16 | 2 |
| wt | $\alpha_{7-330}^{Q20R}$ | 199 ± 46 | 3 |
| wt | $\gamma_{7-323}^{R20Q}$ | 2239 ± 124 | 3 |
| S507D | $\gamma_{7-323}$ | 4706 ± 228 | 2 |
| **RepoMan$_{303–515}$** | | | |
| wt | $\gamma_{7-323}$ | 77 ± 12 | 4 |
| **RepoMan$_{348–450}$** | | | |
| wt | $\gamma_{7-323}$ | 124 ± 6 | 4 |
| T394D | $\gamma_{7-323}$ | 3895 ± 265 | 3 |
| **RepoMan$_{383–441}$** | | | |
| wt | $\gamma_{7-323}$ | 117 ± 10 | 3 |
| **RepoMan$_{383–423}$** | | | |
| wt | $\alpha_{7-330}$ | 778 ± 65 | 4 |
| wt | $\gamma_{7-323}$ | 133 ± 5 | 3 |
| wt | $\gamma_{7-308}$ | 123 ± 24 | 4 |
| wt | $\alpha_{7-330}^{Q20R}$ | 267 ± 28 | 3 |
| wt | $\gamma_{7-323}^{R20Q}$ | 683 ± 91 | 3 |
| wt | $\alpha_{7-330}^{Q20R/R23K}$ | 232 ± 26 | 4 |
| **RepoMan$_{383–404}$** | | | |
| wt | $\gamma_{7-323}$ | 661 ± 160 | 3 |

## The specific recruitment of PP1γ by Ki-67 and RepoMan

How PP1 regulators selectively recruit specific PP1 isoforms to distinct substrates remains an important, open question. PP1 has three isoforms—α, β and γ—that differ primarily in the last ~30 residues of their C-termini; because of this, it has been generally assumed that regulators that bind preferentially to one isoform interact directly with the C-terminus (*Takagi et al., 2014*). In vivo, Ki-67 and RepoMan bind specifically to the β- and γ-isoforms, but not the α-isoform, of PP1 (*Booth et al., 2014*; *Trinkle-Mulcahy et al., 2006*; *Vagnarelli et al., 2011*). As PP1γ is recruited more efficiently than PP1β (*Booth et al., 2014*), we focused our study on the PP1α and PP1γ isoforms. We confirmed this in vitro using ITC, which showed that the affinity of both Ki-67$_{496-536}$ and RepoMan$_{383-423}$ for PP1α$_{7-330}$ is ~four–six-fold weaker than for PP1γ$_{7-323}$ (*Table 1*, *Figure 1—figure supplement 1*). To elucidate the molecular basis of isoform selectivity, we first tested the role of the PP1γ C-terminal residues. Deleting the last 15 residues of PP1γ had no impact on binding (*Table 1*, *Figure 1—figure supplement 1*). We then tested if RepoMan binds PP1α differently by determining the structure of the RepoMan$_{383-441}$:PP1α$_{7-300}$ holoenzyme (2.6 Å; *Figure 1—source data 1*). The structures are identical (backbone RMSD = 0.22 Å), demonstrating that the isoform selectivity is due to one or more of the six amino acid differences in the PP1 catalytic domain (residues 7–300). Only two of these differing residues are located near the Ki-67/RepoMan:PP1 interface, Arg20/Gln20 (γ/α) and Lys23/Arg23 (γ/α; *Figure 2A*). We generated variants of both isoforms in which one or both of these residues were mutated to that of the other (PP1γ$^{R20Q}$, PP1γ$^{R20Q/K23R}$ and PP1α$^{Q20R}$) and determined their affinities for Ki-67$_{496-536}$ and RepoMan$_{383-423}$ using ITC (*Figure 2B*; *Table 1*; *Figure 1—figure supplement 1*). The change of only a single amino acid, R20/Q20, switches PP1γ into a 'PP1α'-like isoform (PP1γ$^{R20Q}$; $K_D$ = 2239 ± 124 nM; ten fold weaker binding) and PP1α into a 'PP1γ'-like isoform (PP1α$^{Q20R}$; $K_D$ = 199 ± 46 nM; similar binding to that of PP1γ). Although R20 does not interact with Ki-67 or RepoMan directly, it confers selectively through its interactions with PP1 which order the L1 loop. Namely, the Arg20 sidechain forms a salt bridge with PP1 residue Glu77 and makes a planar stacking interaction (cation/π interaction) with Phe81. Neither interaction is possible with Q20, as the side chain is both uncharged and too short. Thus, only in PP1γ is this pocket ordered and readily available for binding, which allows for the isoform specific interaction of Ki-67 and RepoMan.

To confirm that PP1α and PP1γ residue 20 defines isoform specificity in vivo, we used immunofluorescence microscopy. Transiently expressed GFP-PP1γ was enriched at anaphase chromosomes (*Figure 2C,D*; *Figure 2—figure supplement 1*). In contrast, GFP-PP1α showed a more diffuse distribution. Changing this single residue in either PP1γ (PP1γ$^{R20Q}$) or PP1α (PP1α$^{Q20R}$) reversed this localization (*Figure 2C,D*; *Figure 2—figure supplement 1*). To confirm that the observed isoform specificity was due to a direct interaction between Ki-67/RepoMan and PP1, we used GFP-Trap and tethering experiments. GFP-traps showed that GFP-PP1γ and GFP-PP1α$^{Q20R}$ bind endogenous Ki-67 in prometaphase-arrested cells, while GFP-PP1α and GFP-PP1γ$^{R20Q}$ bind much more weakly (*Figure 2E*). Similar GFP-traps showed that GFP-PP1γ, GFP-PP1β and GFP-PP1α$^{Q20R}$ bind both endogenous and ectopically expressed RepoMan in prometaphase-arrested cells, while GFP-PP1α, GFP-PP1γ$^{R20Q}$ and GFP-PP1β$^{R19Q}$ bind much more weakly (*Figure 2—figure supplement 1*).

Similarly, tethering experiments showed that a GFP:LacI fusion of Ki-67$_{301-700}$ wt or RepoMan wt results in the recruitment of co-expressed RFP-PP1γ to a LacO array that is integrated at a single locus in DT40 chicken cells. In contrast, RFP-PP1γ$^{R20Q}$ fails to localize, demonstrating it no longer binds Ki-67 and RepoMan (*Figure 2F,G*, *Figure 2—figure supplement 2*). Remarkably, the R20Q mutation abolishes PP1γ localization as effectively as mutating the canonical RVxF motif in RepoMan to RATA. Together, these data reveal that the isoform specificity of Ki-67 and RepoMan is not defined by residues in the PP1 C-terminus, but instead by a single residue in the PP1 catalytic domain, R20/Q20 (γ/α). This is a fundamental result, as it demonstrates that regulator-specific isoform selectively is not only achieved through interactions with the C-terminus, but also through interactions with the structured catalytic domain.

## Dynamic regulation of PP1γ recruitment to chromosomes by Aurora B Kinase

Unlike PP1α, the localization of PP1γ changes dramatically during mitosis, with the bulk of PP1γ relocalizing to chromosomes at anaphase onset. This relocalization is dependent on its ability to

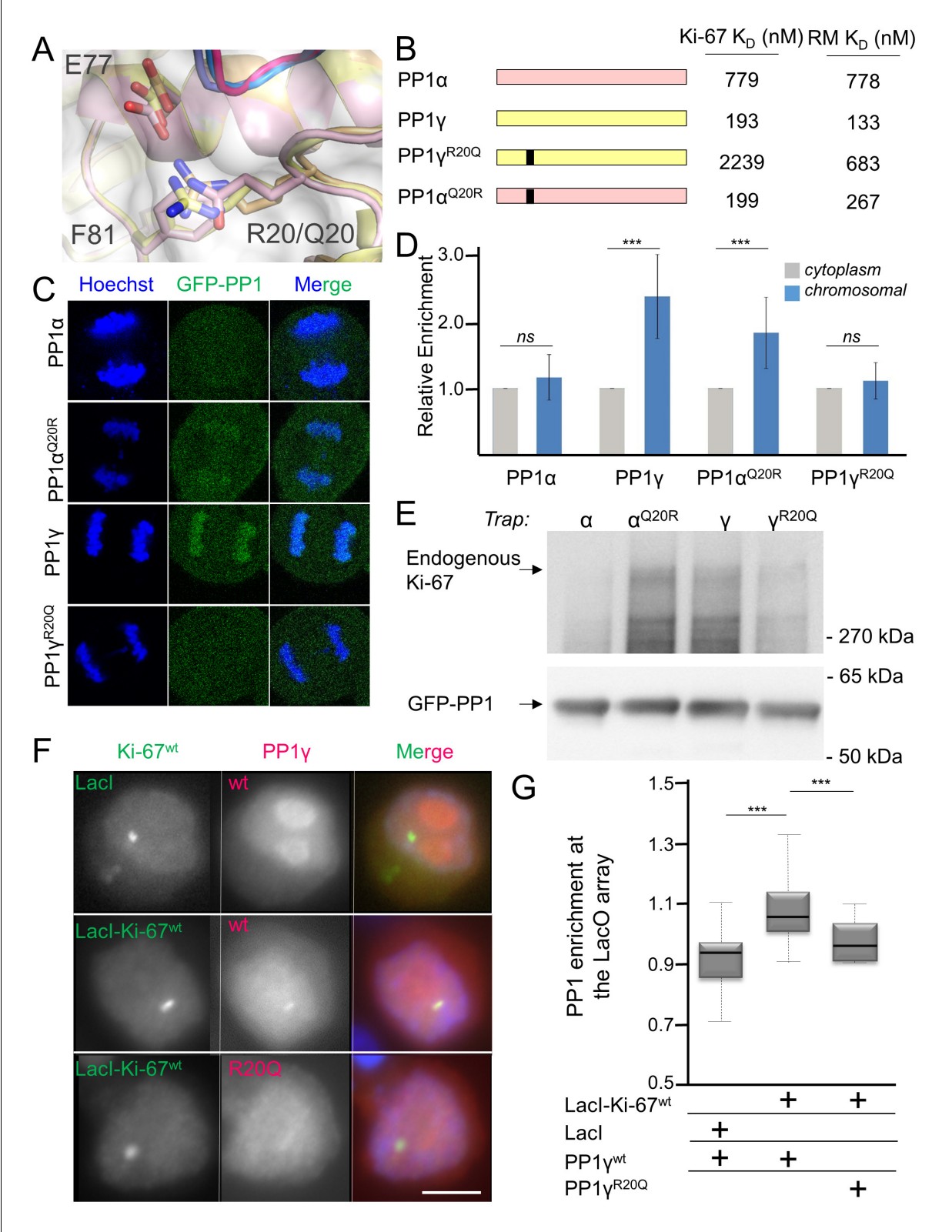

| | | Ki-67 K$_D$ (nM) | RM K$_D$ (nM) |
|---|---|---|---|
| PP1α | | 779 | 778 |
| PP1γ | | 193 | 133 |
| PP1γ$^{R20Q}$ | | 2239 | 683 |
| PP1α$^{Q20R}$ | | 199 | 267 |

**Figure 2.** Ki-67/RepoMan isoform specificity is defined by PP1 residue 20. (**A**) Overlay of PP1 from the Ki-67$_{496-536}$:PP1γ$_{7-308}$ (magenta:yellow), RepoMan$_{383-423}$:PP1γ$_{7-308}$ (blue:orange) and RepoMan$_{383-423}$:PP1α$_{7-300}$ (lavender:pink) complexes. R20 (PP1γ), Q20 (PP1α), E77 (PP1α/PP1γ) and F81 (PP1α/PP1γ) are shown as sticks. (**B**) Cartoon illustrating the PP1 variants generated for this study; colored as in A. The resulting K$_D$ values of Ki-67 and RepoMan titrated with the different PP1 variants are shown. (**C**) Subcellular distributions of GFP-PP1 fusions. HeLa cells were transfected with the

*Figure 2 continued on next page*

*Figure 2 continued*

indicated GFP-PP1 variants. The green fluorescence was visualized by confocal microscopy. Anaphase chromosomes were detected using Hoechst and live imaging. (D) Quantification of the relative enrichment of PP1 variants on chromosomes in C. Average and standard deviation are shown (T-test using Welch's correction). (E) HEK293T cells were transfected with GFP-PP1 variants. The micrococcal-nuclease-treated cell lysates from nocodazole arrested cells were used for GFP trapping. The traps were processed using immunoblotting. (F) Chicken DT40 cells carrying a LacO array inserted in a single locus were transfected with GFP:LacI or GFP:LacI:Ki-67$_{301-700}^{wt}$ constructs (green) and with RFP:PP1$^{wt}$ or RFP:PP1$^{R20Q}$ (red). GFP:LacI:Ki-67$_{301-700}^{wt}$ and RFP:PP1$^{wt}$ both accumulate at the LacO array, however the RFP:PP1$^{R20Q}$ fails to accumulate together with GFP:LacI:Ki-67$^{wt}$ at the locus. Scale bar 5 µm. (G) Quantification of the enrichment of PP1 at the locus from the experiment in (F) (Mann-Whitney test) between: 1) GFP:LacI/RFP:PP1$^{wt}$ and GFP:LacI:Ki-67$^{wt}$/RFP:PP1$^{wt}$ or 2) GFP:LacI:Ki-67$^{wt}$/RFP:PP1$^{wt}$ and GFP:LacI:Ki-67$^{wt}$/RFP:PP1$^{R20Q}$.

The following figure supplements are available for figure 2:

**Figure supplement 1.** RepoMan and Ki-67 isoform specificity is defined by PP1 residue 20 throughout the cell cycle.

**Figure supplement 2.** PP1 residue 20 is also critical for tethering by RepoMan.

bind effectively to Ki-67 and RepoMan (*Figure 2C*, *Figure 2—figure supplement 2*) (*Trinkle-Mulcahy et al., 2003*; *Trinkle-Mulcahy et al., 2001*) and is regulated by phosphorylation (*Figure 3A*) (*Vagnarelli et al., 2006*; *Qian et al., 2013*; *Vagnarelli et al., 2011*; *Qian et al., 2015*). For example, RepoMan S400, T412 and T419 are phosphorylated by CDK1-cyclin B (*Vagnarelli et al., 2011*; *Qian et al., 2015*). Mutating these residues to phosphomimetics inhibits PP1γ binding, as evidenced by the inability of EGFP-RepoMan$^{3D}$ to pull-down PP1γ from non-synchronized HEK293T cells [see *Figure 3F* in *Qian et al. (2015)*] and an inability of RFP-PP1γ to localize to the LacO locus when co-expressed with GFP:LacI-RepoMan$^{3D}$ (*Figure 3B*). Notably, the identity of the kinase that phosphorylates the S507$_{Ki-67}$ and T394$_{RM}$ sites, both of which have been identified in proteomic screens (*Dephoure et al., 2008*; *Nousiainen et al., 2006*), has remained elusive. Because Aurora B kinase is redistributed from centromeres to the spindle midzone at anaphase onset and because the S507$_{Ki-67}$ and T394$_{RM}$ sequences match canonical Aurora B kinase phosphorylation motifs (*Kettenbach et al., 2011*), we reasoned that these residues are phosphorylated by Aurora B kinase. Using in vitro phosphorylation assays coupled with NMR spectroscopy and mass spectrometry, we showed that both S507$_{Ki-67}$ and T394$_{RM}$ (*Figure 3C*, *Figure 3—figure supplement 1*) are phosphorylated directly by Aurora B kinase. Furthermore, mutating S507$_{Ki-67}$ and T394$_{RM}$ to a phosphorylation mimetic (Ki-67$_{496-536}^{S507D}$/RepoMan$_{348-450}^{T394D}$) profoundly weakens their interactions with PP1 (~25-30-fold reduction in affinity; *Table 1*). Finally, the expression of GFP-LacI-RepoMan$^{T394D}$ results in a significant reduction of the recruitment of RFP-PP1γ to the LacO locus (*Figure 3B*), i.e., to a level that is nearly identical to that observed for GFP-LacI-RepoMan$^{RATA}$ and GFP-LacI-RepoMan$^{3D}$. These data demonstrate that phosphorylation by Aurora B kinase in (pro) metaphase inhibits PP1γ binding to Ki-67 and RepoMan and, as a consequence, significantly contributes to prevent their premature recruitment to chromosomes.

## Conclusion

Our study builds on previous work to reveal how two mitotic exit phosphatases (Ki-67:PP1 and RepoMan:PP1) are assembled and recruited to their cellular targets, how they selectively bind the γ-isoform of PP1 and how the assembly of these holoenzymes is controlled by Aurora B kinase phosphorylation. These are key advances as, until this work, there was essentially no molecular data on mitotic phosphatase assembly and function. Our data now explains why the phosphorylation of RepoMan at three distinct sites (S400, S412 and S419) by Cdk1 inhibits PP1 binding (*Qian et al., 2015*). Namely, it is similar to the mechanism by which Aurora B kinase inhibits Ki-67:PP1 and RepoMan:PP1 holoenzyme formation. All three residues, like the Aurora B targets S507$_{Ki-67}$ and T394$_{RM}$, are part of the PP1 interaction motif and their phosphorylation blocks the interaction with PP1, which, in turn, prevents the premature targeting of PP1 to chromatin. This regulation of holoenzyme assembly is critical as the premature targeting of PP1 to chromosomes leads to an increase in chromosome misalignment and a weakened spindle assembly checkpoint (*Qian et al., 2015*). In contrast, the expression of a PP1-binding mutant of RepoMan in HeLa cells results in extensive cell death,

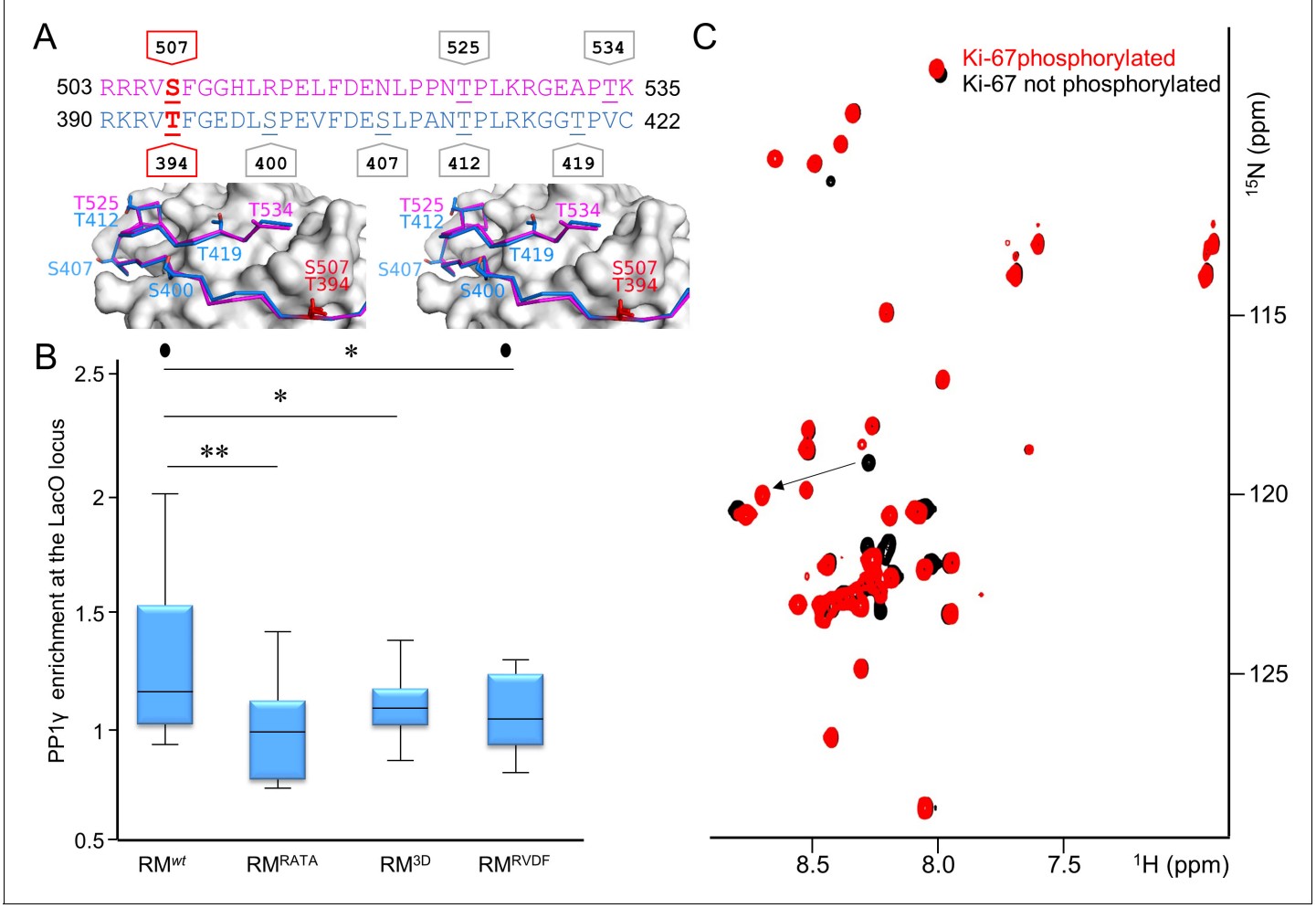

**Figure 3.** Aurora B kinase phosphorylates Ki-67 S507 and RepoMan T394 to inhibit holoenzyme formation. (**A**) Stereo image of Ki-67 (magenta) and RepoMan (blue) bound to PP1 (grey) with serine and threonine residues shown as sticks and labeled. The 'x' residues of the RVxF motifs, $S507_{Ki-67}$ and $T394_{RM}$, are shown and highlighted in red. (**B**) Chicken DT40 cells carrying a LacO array inserted in a single locus were transfected with different GFP: LacI:RepoMan constructs (green) and with RFP:PP1 (red) and the enrichment of PP1 at the locus was calculated. Both the CDK-1 (3D) and the Aurora B (RVDF) phosphomimetic mutants cause a significant decrease in PP1 accumulation at the locus although less pronounced than the RATA mutant (Mann-Whitney test). (**C**) Overlay of the 2D [$^1$H,$^{15}$N] HSQC spectrum of Ki-67$_{496-536}$(black) and Aurora B kinase phosphorylated Ki-67$_{496-536}$(red). Peaks that correspond to S507 (black) and phosphorylated pS507 (red) are shown by an arrow.

The following figure supplement is available for figure 3:

**Figure supplement 1.** RepoMan RVxF residue T394 is specifically phosphorylated by Aurora B kinase.

---

demonstrating the importance of the RepoMan-PP1 interaction for cell viability (*Trinkle-Mulcahy et al., 2006*).

This work also expands the diversity of SLiMs used by PP1 regulators to bind PP1 and demonstrates a novel mechanism by which isoform selectivity is achieved in PP1 holoenzymes, a question that has been under investigation for the last 20 years. That is, Ki-67 and RepoMan selectively assemble with PP1γ because the KiR-SLiM distinguishes between a single residue in PP1: $Arg20_{PP1\gamma}$ and $Gln20_{PP1\alpha}$. Finally, since the KiR-SLiM is present in just two regulators, the newly discovered KiR-SLiM binding pocket defines a novel surface on PP1 that is targetable for the development of drugs that inhibit only 1% of the distinct PP1 holoenzymes in the cell (2 of ~200) (*Hendrickx et al., 2009*). This is of profound interest because both Ki-67 and RepoMan are highly upregulated in multiple cancers (*Dowsett et al., 2011*; *Vagnarelli, 2014*). Targeting unique regulator binding pockets is a

powerful approach as we recently discovered that the newly discovered LxVP SLiM substrate interaction surface is a well-known drug target site in the ser/thr phosphatase Calcineurin (*Grigoriu et al., 2013*). Given the diversity of unique interactions that are now being revealed between PSPs and their plethora of regulators, it is now clear that these novel, unique protein:protein interaction sites, especially that defined by the KiR-SLiM, provide immediate opportunities for the design of novel, highly specific therapeutics.

## Materials and methods

### Cloning and expression

The coding sequences of human Ki-67$_{496-536}$, RepoMan$_{303-515}$, RepoMan$_{348-450}$, RepoMan$_{383-441}$ and RepoMan$_{383-423}$ were sub-cloned from pGEX4-T3-Ki-67 (*Booth et al., 2014*) and pEGFP-Repoman (*Qian et al., 2013*) into the pET-M30-MBP vector containing a N-terminal his$_6$-tag followed by maltose binding protein (MBP) and a TEV (tobacco etch virus) protease cleavage site. *Escherichia coli* BL21 (DE3) RPIL cells (Agilent; Lexington, MA) transformed with the expression vector for Ki-67 (or RepoMan) were grown at 37°C in LB broth containing selective antibiotics. The proteins were overexpressed by the addition of 1 mM isopropylthio-β-D-galactoside (IPTG) when the optical density (OD$_{600}$) reached 1.2 and the cultures were grown for an additional 18–20 hr at 16°C. Cells were harvested by centrifugation (6000 x$g$, 12 min, 4°C) and stored at −80°C until purification. Ki-67$^{S507D}$ and RepoMan$^{T394D}$ mutant was generated using the QuikChange site-directed mutagenesis kit (Agilent). The RepoMan$_{383-404}$ peptide was synthesized and HPLC purified (>95% purity; Biosynthesis, Inc.; Lewisville, TX). PP1γ$_{7-323}$$^{R20Q}$, PP1γ$_{7-323}$$^{R20QR23K}$ and PP1α$_{7-330}$$^{Q20R}$ were generated using the QuikChange site-directed mutagenesis kit (Agilent). Uniformly $^{15}$N and $^{15}$N/$^{13}$C labeled RepoMan$_{348-450}$ and $^{15}$N labeled Ki-67$_{496-536}$ was produced using the same procedure except that the cells were grown in M9 minimal medium supplemented with $^{15}$N ammonium chloride (1 g/L) and/or $^{13}$C-D-glucose (4 g/L) as the sole nitrogen and carbon sources, respectively. Cloning, expression and purification of PP1α$_{7-330}$, PP1α$_{7-300}$ and PP1γ$_{7-323}$ was performed as previously described (*Choy et al., 2014*; *O'Connell et al., 2012*). Cloning, expression and purification of PP1γ$_{7-308}$ was performed following the methods described in *Choy et al. (2014)*.

### Protein purification

*E. coli* cell pellets containing Ki-67 (or RepoMan) were resuspended in ice-cold lysis buffer (50 mM Tris pH 8.0, 0.5 M NaCl, 5 mM imidazole, 0.1% Triton X-100 containing EDTA-free protease inhibitor tablet [SigmaAldrich; St. Louis, MO]), lysed by high-pressure cell homogenization (Avestin C3 Emulsiflex; Canada) and centrifuged (35,000 x$g$, 40 min, 4°C). The supernatant was loaded onto a HisTrap HP column (GE Healthcare; Boston, MA) pre-equilibrated with 50 mM Tris pH 8.0, 500 mM NaCl and 5 mM imidazole (Buffer A) and was eluted using a linear gradient of Buffer B (50 mM Tris pH 8.0, 500 mM NaCl, 500 mM imidazole). Fractions containing the protein were pooled and dialyzed overnight at 4°C (50 mM Tris pH 8.0, 500 mM NaCl) with TEV protease to cleave the His$_6$-MBP tag. The cleaved protein was incubated with Ni$^{2+}$-NTA beads (GE Healthcare) and the flow-through collected. The protein was then heat purified at 95°C (600 rpm, 15 min), the supernatant collected and concentrated and, in a final step, purified using size exclusion chromatography (SEC; Superdex 75 26/60 [GE Healthcare]) pre-equilibrated in ITC Buffer (20 mM Tris pH 8, 500 mM NaCl, 0.5 mM TCEP, 1 mM MnCl$_2$) or NMR Buffer (20 mM Tris-acetate pH 6.5, 150 mM NaCl, 0.5 mM TCEP). Fractions were pooled, concentrated and stored at −20°C. RepoMan constructs were purified identically.

To purify the Ki-67$_{496-536}$:PP1γ$_{7-308}$, RepoMan$_{383-423}$:PP1γ$_{7-308}$ and RepoMan$_{383-441}$:PP1α$_{7-300}$ holoenzyme complexes, a cell pellet expressing PP1γ$_{7-308}$ was lysed in PP1 Lysis Buffer (25 mM Tris pH 8.0, 700 mM NaCl, 5 mM imidazole, 1 mM MnCl$_2$, 0.01% Triton X-100), clarified by ultracentrifugation and immobilized on Ni$^{2+}$-NTA resin. Bound His$_6$-PP1 was washed with PP1 Buffer A (25 mM Tris pH 8.0, 700 mM NaCl, 5 mM imidazole, 1 mM MnCl$_2$), followed with a stringent wash containing 10% PP1 Buffer B (25 mM Tris pH 8.0, 700 mM NaCl, 250 mM imidazole, 1 mM MnCl$_2$) at 4°C. The protein was eluted using 100% PP1 Buffer B and purified using SEC pre-equilibrated in ITC Buffer (20 mM Tris pH 8, 500 mM NaCl, 0.5 mM TCEP, 1 mM MnCl$_2$). Peak fractions were incubated overnight with TEV protease at 4°C. The cleaved protein was incubated with Ni$^{2+}$-NTA beads (GE Healthcare) and the flow-through collected. The flow-through was combined with excess Ki-

$67_{496\text{-}536}$ (or RepoMan$_{383\text{-}423}$, RepoMan$_{383\text{-}441}$) concentrated and the complex purified using SEC (pre-equilibrated in crystallization buffer: 20 mM Tris pH 8, 500 mM NaCl, 0.5 mM TCEP). Fractions containing the holoenzyme complex were concentrated (RepoMan$_{383\text{-}423}$:PP1$\gamma_{7-308}$, 8.4 mg/mL; Ki-67$_{496\text{-}536}$:PP1$\gamma_{7-308}$ 6.4 mg/ml) and subsequently used for crystallization trials.

## NMR measurements

All NMR spectra were recorded on a Bruker Avance II 500 spectrometer equipped with a TCI HCN z-gradient cryoprobe (298 K). NMR samples were prepared in NMR buffer (20 mM Tris-acetate pH 6.5, 150 mM NaCl, 0.5 mM TCEP) containing 10% (v/v) $D_2O$. The sequence-specific backbone assignment for RepoMan$_{348\text{-}450}$ (0.25 mM concentration) was obtained by analyzing 2D [$^1$H,$^{15}$N] HSQC, 3D HNCA, 3D HN(CO)CA, 3D HNCACB, 3D CBCA(CO)NH and 3D (H)CC(CO)NH ($\tau_m$= 12 ms) spectra. All spectra were processed using NMRPipe (*Delaglio et al., 1995*) and analyzed using SPARKY (*Goddard and Kneller, 2004*).

## Crystallization and structure determination

The Ki-67$_{496\text{-}536}$:PP1$\gamma_{7-308}$ holoenzyme crystallized in 1.9 M Sodium Malonate pH 4.0 (hanging drop vapor diffusion at 4°C). Crystals were cryo-protected by a 60 s soak in mother liquor supplemented with 40% glycerol and immediately flash frozen. Data for the Ki-67$_{496\text{-}536}$:PP1$\gamma_{7-308}$ holoenzyme crystal structure were collected to 2.0 Å at beamline 12.2 at the Stanford Synchrotron Radiation Lightsource (SSRL) at 100 K and a wavelength of 0.98 Å using a Pilatus 6M PAD detector. The Ki-67$_{496\text{-}536}$:PP1$\gamma_{7-308}$ crystal data was processed using XDS (*Kabsch, 2010*), Aimless (*Evans and Murshudov, 2013*) and Truncate (*French and Wilson, 1978*). The data was analyzed using *phenix.xtriage* and the intensity statistics suggested merohedral twinning (twin fraction of 0.48) with the space group of P6$_1$ and twin law h, -h-k, -l. The structure was solved by molecular replacement using Phaser (*McCoy et al., 2007*) as implemented in PHENIX (*Zwart et al., 2008*) (PDBID 5INB was used as the search model). The model was completed using iterative rounds of refinement in PHENIX (*McCoy et al., 2007*) and manual building using Coot (*Emsley et al., 2010*) (Ramachandran statistics: 95.05% favored, 4.95% allowed). The final structure was refined in PHENIX with the twin law. The RepoMan$_{383\text{-}423}$:PP1$\gamma_{7-308}$ complex crystallized in 1 M Sodium Malonate pH 4.3 (sitting drop vapor diffusion method at 4°C). Crystals were cryo-protected by a 30 s soak in 1 M Sodium Malonate pH 4 supplemented with 40% glycerol and immediately flash frozen. Data for the RepoMan$_{383\text{-}423}$: PP1$\gamma_{7-308}$ holoenzyme crystal structure were collected to 1.3 Å at the beamline 12.2 at Stanford Synchrotron Radiation Lightsource (SSRL) at 100 K and a wavelength of 0.98 Å using a Pilatus 6M PAD detector. The RepoMan$_{383\text{-}423}$:PP1$\gamma_{7-308}$ crystal data were processed using XDS (*Kabsch, 2010*), Aimless (*Evans and Murshudov, 2013*) and Truncate (*French and Wilson, 1978*). The structure was solved by molecular replacement using Phaser (*McCoy et al., 2007*) as implemented in PHENIX (*Zwart et al., 2008*) (PDB ID 1JK7 (*Maynes et al., 2001*) was used as the search model). A solution was obtained in space group P6$_1$22. The model was completed using iterative rounds of refinement in PHENIX and manual building using Coot (*Emsley et al., 2010*) (Ramachandran statistics: 96.3% favored, 3.7% allowed). The RepoMan$_{383\text{-}441}$:PP1$\alpha_{7-300}$ holoenzyme crystallized in 100 mM Sodium Malonate pH 4.0, 12% PEG 3350. Crystals were cryo-protected by a 30 s soak in mother liquor supplemented with 30% glycerol and immediately flash frozen. Data were collected to 2.6 Å at the National Synchrotron Light Source (BNL) Beamline X25 at 100 K and a wavelength of 1.1 Å using Dectris pilatus 6M detector. Data were indexed, scaled and merged using HKL2000 0.98.692i (*Otwinowski and Minor, 1997*). The structure was solved by molecular replacement using Phaser as implemented in PHENIX (PDB ID 4MOV was used as the search model (*Choy et al., 2014*; *McCoy et al., 2007*; *Zwart et al., 2008*). A solution was obtained in space group P2$_1$2$_1$2$_1$. The model was completed using iterative rounds of refinement in PHENIX (*McCoy et al., 2007*) and manual building using Coot (*Emsley et al., 2010*) (Ramachandran statistics: 95% favored, 5% allowed).

## Isothermal titration calorimetry

His$_6$-tagged-PP1 constructs (PP1$\alpha_{7-330}$, PP1$\gamma_{7-308}$, PP1$\gamma_{7-323}$, PP1$\gamma_{7-323}$$^{R20Q}$, PP1$\gamma_{7-323}$$^{R20QR23K}$ and PP1$\alpha_{7-330}$$^{Q20R}$) used for ITC were purified as follows. PP1 was lysed and purified using Ni$^{2+}$-affinity chromatography and SEC (pre-equilibrated ITC buffer, 20 mM Tris pH 8, 500 mM

NaCl, 0.5 mM TCEP, 1 mM MnCl$_2$). Ki-67 or RepoMan (30 µM to 40 µM) was titrated into PP1 (3 µM to 4 µM) using a VP-ITC micro-calorimeter at 25°C (Malvern; United Kingdom). Data were analyzed using NITPIC, SEDPHAT and GUSSI (*Scheuermann and Brautigam, 2015*; *Zhao et al., 2015*).

## In vitro phosphorylation

Aurora Kinase B (AuKB) was expressed and purified as a GST-tagged protein as previously described (*Qian et al., 2013*). In-vitro phosphorylation of Ki-67$_{496-536}$ or RepoMan$_{348-450}$ was achieved by incubation with AuKB at a molar ratio 20:1 in phosphorylation buffer (20 mM HEPES pH 7.5, 0.5 mM EDTA, 2 mM DTT, 20 mM MgCl$_2$ and 10 mM ATP). The reaction was allowed to proceed for 16 hr at 30°C. Following incubation, the protein was concentrated and purified using SEC. The phosphorylation of both Ki-67$_{496-536}$ and RepoMan$_{348-450}$ was confirmed using ESI-MS.

## RepoSLiM identification

HMMER (*Finn et al., 2015*) (using the rp75 UniProt database) identified proteins with sequences similar to the RepoMan PP1-binding domain. 107 sequences were identified, spanning a diversity of species, from human to Xenopus. WebLogo was used to generate the resulting logo. The UniProtKB/Swiss-Prot database was then scanned using ScanProsite (*de Castro et al., 2006*) using the pattern F-D-x-x-[LMK]-P-[PA]-[NSDAI]-[TSA]-P-[LIV]-[RKQC]-[RK]-G-x-[TSIAL]-[PS] to identify proteins that contain this motif, which resulted in the identification of only Ki-67 and RepoMan proteins.

## Cell culture and RNA interference

HeLa Kyoto cells were maintained in DMEM supplemented with 10% FBS. DT40 cells carrying a single integration of the LacO array (*Vagnarelli et al., 2006*) were cultured in RPMI1640 supplemented with 10% FBS and 1% chicken serum. For RNAi treatments, HeLa cells in exponential growth were seeded in 6 well plates with polylysine-coated glass coverslips and grown overnight. Transfections were performed using Polyplus jetPRIME (PEQLAB; Germany) with the indicated siRNA oligos and analyzed 48 hr later as previously described after 3 hr nocodaole arrest (*Vagnarelli et al., 2006*). For the rescue experiments HeLa cells at 50% confluence were transfected with 400 ng of plasmid DNA and 50 nM of siRNA oligonucleotides and analyzed 48 hr post-transfection after 3 hr nocodaole arrest. Transient transfections for DT40 were conducted as previously described (*Qian et al., 2013*). For quantification of the enrichment at the Laci locus, cells were fixed with paraformaldehyde 24 hr after transfection.

## Live imaging

HeLa cells were seeded in a 4-Chamber 35 mm Glass Bottom Dish with 20 mm microwell, #1 cover glass (Cellvis) and transfected with X-tremeGENE 9 DNA Transfection Reagent (Roche) according to manufacturer's protocol. The DNA was stained with Hoechst 33,342 (Tocris; United Kingdom) for live imaging. Confocal images were acquired with a Leica TCS SPE laser-scanning confocal system mounted on a Leica DMI 4000B microscope, equipped with a Leica ACS APO 63X 1.30NA oil DIC objective and a live-imaging chamber ensuring 37°C and 5% CO$_2$.

## Indirect immunofluorescence and microscopy analyses

Cells were fixed in 4% PFA and processed as previously described (*Rosenberg et al., 2011*). 3D data sets were acquired using a cooled CCD camera (CoolSNAP HQ2 firewire) on a wide-field microscope (Eclipse Ti, NIKON) with a NA 1.45 Plan Apochromat lens. The data sets were deconvolved with NIS-Element AR (NIKON). Three-dimensional data sets were converted to MIP in NIS-Element AR, exported as TIFF files, and imported into Adobe Photoshop for final presentation.

## Mutagenesis

RepoMan mutants were generated by GeneArt Site-Directed Mutagenesis system (Thermo Fisher Scientific/Invitrogen; Waltham, Ma) using the plasmids GFP:RepoMan WT and GFP:RepoMan$^{T412A,}$ $^{T419A}$ and GFP:Laci:RepoMan. The following primer sequences were used:

*Oligo resistant mutant*:5'AAAGAGTCCGAGATGACTGACTAGTCCTGAAAGGAAGGTCTCAGCG 3'

*RepoMan 3A mutant*: 5'TTGGAGAGGACTTAGCCCCGGAAGTGTTTGA3' on GFP:RepoMan$^{T412A, T419A}$ oligo-resistant.

*RepoMan F404A mutant*: 5'TAAGCCCGGAAGTGGCTGATGAATCTTTGCC3'

*GFP:RepoMan 3D*: 5'TTGGAGAGGACTTAGACCCGGAAGTGTTTGA3'

*GFP:Laci:RepoMan$^{3D}$* was generated by cloning the Laci sequence into XhoI/KpnI of GFP:RepoMan$^{3D}$

*GFP:Laci:RepoMan$^{RVDF}$ (T394D)*: 5'AAGAGGAAGAGAGTTGACTTTGGAGAGGACTTA3' on GFP:LAci:RepoMan

*RFP:PP1$\gamma$R20Q*: 5'CGGCTGCTGGAGGTGCAAGGATCAAAACCAGGT3'

## Quantification of the chromatin enrichment (prometaphase or anaphase)

Images of prometaphases (or anaphase for the PP1$\gamma$ experiment) from transfected cells were acquired, the 3D stacks were deconvolved and projected using MIP. A 10 × 10 pixel area contained within the chromosomes was used to measure the total intensity of the signal. Three different measurements per mitosis on different chromosomes and 3 different cytoplasmic areas were collected and averaged. An area of the same size outside the cells was used to identify the background signal in each image, and this value was subtracted from the measurement of the chromosome and cytoplasmic area. The GFP (or RFP for PP1$\gamma$) intensity on the chromosomes was then normalized against the cytoplasmic intensity. For quantification of PP1-binding at the LacO locus, images of interphase transfected cells were acquired and the intensity of PP1 staining at the GFP spot was calculated relative to the average nuclear intensity. The 3-dimensional data sets obtained at the same exposure were projected as mean intensities. A 12 × 12 pixel area containing the GFP spot was used to measure the total intensity of the signal. An area of the same size was used to identify the background signal in each cell, and this value was subtracted from the measurement of the nuclear and spot area. The data were analyzed using the Mann-Whitney U test.

## Accession numbers

All NMR chemical shifts were deposited in the BioMagResBank (BMRB 25981). Atomic coordinates and structure factors have been deposited in the Protein Data Bank (Ki-67:PP1$\gamma$, PDBID 5J28; RepoMan: PP1$\gamma$, PDBID 5INB; RepoMan:PP1$\alpha$, PDBID 5IOH).

# Acknowledgements

The authors thank Dr. R Bajaj for providing Aurora B kinase and Mr. Haejun Cho for help with Ki-67 expression and purification. This research is based in part on data obtained at the Brown University Structural Biology Core Facility, which is supported by the Division of Biology and Medicine, Brown University. Crystallographic data were collected on beamline 12.2 at the Stanford Synchrotron Radiation Lightsource and X25 at the National Synchrotron Light Source. Use of the Stanford Synchrotron Radiation Lightsource, SLAC National Accelerator Laboratory, is supported by the U.S. Department of Energy, Office of Science, Office of Basic Energy Sciences under Contract No. DE-AC02-76SF00515. The SSRL Structural Molecular Biology Program is supported by the DOE Office of Biological and Environmental Research, and by the National Institutes of Health, National Institute of General Medical Sciences (including P41GM103393). The National Synchrotron Light Source is supported principally by the Offices of Biological and Environmental Research and of Basic Energy Sciences of the United States Department of Energy and by the National Center for Research Resources of the National Institutes of Health. The contents of this publication are solely the responsibility of the authors and do not necessarily represent the official views of NIGMS or NIH.

## Additional information

### Funding

| Funder | Grant reference number | Author |
| --- | --- | --- |
| Fonds Wetenschappelijk Onderzoek | Flanders - G.0473.12 | Mathieu Bollen |

| Fonds Wetenschappelijk Onderzoek | Flanders - G.0482.12 | Mathieu Bollen |
|---|---|---|
| Flemish Concerted Research Action | GOA 15/016 | Mathieu Bollen |
| Biotechnology and Biological Sciences Research Council | BB/K017632/1 | Paola Vagnarelli |
| National Institute of Neurological Disorders and Stroke | R01NS091336 | Wolfgang Peti |
| National Institute of General Medical Sciences | R01GM098482 | Rebecca Page |

The funders had no role in study design, data collection and interpretation, or the decision to submit the work for publication.

## Author contributions

GSK, MB, PV, WP, Conception and design, Acquisition of data, Analysis and interpretation of data, Drafting or revising the article; EG, Acquisition of data, Analysis and interpretation of data; SDM, Acquisition of data, analysis and interpretation of data; RP, Conception and design, acquisition of data, analysis and interpretation of data, drafting or revising the article, all authors discussed the data and manuscript.

## Author ORCIDs

Paola Vagnarelli, http://orcid.org/0000-0002-0000-2271
Rebecca Page, http://orcid.org/0000-0002-4645-1232

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
