## [Decision Letter]

[Editors’ note: this article was originally rejected after discussions between the reviewers, but the authors were invited to resubmit after an appeal against the decision.]

Thank you for choosing to send your work entitled "Molecular Assembly of the Ki-67 and RepoMan Mitotic Phosphatases: Regulators of Chromatin Organization and Structure" for consideration at *eLife*. Your initial submission has been assessed by Tony Hunter in consultation with three members of the Board of Reviewing Editors. Although the work is of interest, we are not convinced that the findings presented have the potential significance that we require for publication in *eLife*.

The main linkage between this paper and the original paper in *eLife*, which justifies this as a Research Advance, is the crystal structure of the Ki-67 motif bound to PPI. This new structure is very similar to that of the homologous RepoMan motif bound to PP1γ, which would have been predicted based on the very strong similarity of the PP1-binding motifs of these proteins. In the rest of the manuscript the authors used the RepoMan:PP1 interaction as a proxy for the Ki-67:PP1 interaction, but without validating any of their conclusions on Ki-67 itself, which makes the link to the original *eLife* paper rather tenuous. The Ki-67 and RepoMan motif-PPI structures are quite interesting because the peptide forms a β-hairpin, with its N-terminal part being positioned on the usual RVXF binding site of PP1, but with its C-terminal part turning around 180 degrees, in contrast to other PP1 structures where the polypeptide chain continues more or less straight on the PP1 surface, often establishing a second main contact at the SILK binding site. However, whether Ki67 and RepoMan are the only ligands for this new binding site on PP1, as the authors claim, is not clear, as it may well be that motifs with divergent or even unrelated sequences bind at the same site. Regulation of RV(S/T)F sites by Aurora B kinase has already been established (most notably, for the N-terminal region of Knl1/CASC5), as has the control by Aurora B of RepoMan recruitment to chromatin. Moreover, no direct evidence of the functional importance of the reported RepoMan and Ki-67 interactions with PP1γ for progression through mitosis is provided, for instance as assessed by expressing appropriate mutants in knockdown cells. Taken together, the three editors who reviewed this paper are not convinced that these studies constitute a sufficiently substantial advance on Booth et al. to justify publication as a Research Advance in *eLife*.

We return a high proportion of articles to authors without passing them on for in-depth peer review so that they can be promptly submitted elsewhere. This is not meant as a criticism of the quality of the data or the rigor of the science, but merely reflects our desire to publish only the most influential research. We wish you good luck with your work and we hope you will consider *eLife* for future submissions.

[Editors’ note: what now follows is the decision letter after the authors submitted for further consideration.]

Thank you for submitting your article "Molecular Assembly of the Ki-67 and RepoMan Mitotic Phosphatases: Regulators of Chromatin Organization and Structure" for consideration by *eLife*. Your article has been reviewed by three peer reviewers, one of whom is a member of our Board of Reviewing Editors, and the evaluation has been overseen Tony Hunter as the Senior Editor. The reviewers have opted to remain anonymous.

The reviewers have discussed the reviews with one another and the Reviewing Editor has drafted this decision to help you prepare a revised submission.

Summary:

This Research Advance by Kumar et al. defines the mechanism by which RepoMan (RM) and Ki-67 interact with the PP1γ catalytic subunit. The authors use crystallography, ITC and cell imaging to characterize these interactions. The two main findings are that: (i) RM and Ki-67 interact with PP1γ through a common structural mechanism involving an extended structure that includes the RVxF SLiM, the ΦΦ SLiM and a novel motif they refer to as the KiR SLiM. (ii) The authors define a structural mechanism for the selectivity of RM and Ki-67 for the PP1gamma isoform relative to PP1α as being due to Arg 20 in PP1γ compared to Gln20 in PP1α. By substituting Arg for Gln, the authors convert PP1α to PP1γ -like and vice versa. These results are of interest and warrant publication in *eLife*, subject to the following important corrections and clarifications.

Essential revisions:

Reviewer 1:

Second paragraph of Introduction section: A recent paper in Nature sheds light on an important function of Ki-67 and the authors might elect to cite it.

Sixth paragraph of Results section: "[…]pull down PP1γ[…]": are the data shown anywhere? They are not in Figure 3

The discussion on binding specificity of RM and Ki-67 for isoforms of PP1 is limited to the α and γ subunits, but the authors do not mention the β subunit anywhere. This point should be discussed.

Fifth paragraph of Results section: "GFP-Traps showed that GFP- PP1γ and GFP-PP1αQ20R bind […] in prometaphase-arrested cells". This is an important point. If the authors' theory on phosphoregulation of the interaction is correct, there ought to be no binding in prometaphase. Please clarify.

Is Panel 3C cited anywhere in the text?

First paragraph of Conclusion section: There is additional prior art on the role of PP1 in the control of mitotic checkpoint function, in particular the paper from the Funabiki laboratory (Rosenberg et al. Current Biology 21:942). Please cite it.

Panel 2F: The upper left panel is probably not the WT Ki-67 sequence, but the positive control GFP-LacI. In panel G, the authors could clearly indicate that they are comparing GFP-LacI to GFP-LacI-Ki67.

Please add molecular weight markers to all gels.

Reviewer 3:

1) The statement beginning in the first paragraph of the Results section. ' […] this conserved region[…] constitutes the full PP1 interaction domain'. I could not find evidence for this statement.

2) Also in the same sentence, given the data presented at this point, the comment that 'Ki-67 and RM interact with PP1 using identical mechanisms' is unsubstantiated. Their structural data presented later does give credence to this view.

3) Figure 1 should indicate the Kd and the name of the interaction proteins.

4) Figure 1 should also indicate the Kd and the name of the interaction proteins and also indicate what ΔKiR-SLiM actually is.

5) Third paragraph of Results. The authors identify a new SLiM using HMMER. Did the authors test the role of consensus residues in this motif, for example F517/F404, P523/A416?

6) The authors suggest that Arg20 of PP1γ confers selectivity for RM and Ki-67. The equivalent residue in Gln in PP1α. Unfortunately neither the structure shown in Figure 2 nor the text explains how Arg20 confers this selectivity. What residues on RM and Ki-67 does Arg20 interact with? Are these conserved between RM and Ki-67?

7) The labels in Figure 3 aren't clear.

---

## [Author Response]

[Editors’ note: the author responses to the first round of peer review follow.]

*Thank you for choosing to send your work entitled "Molecular Assembly of the Ki-67 and RepoMan Mitotic Phosphatases: Regulators of Chromatin Organization and Structure" for consideration at eLife. Your initial submission has been assessed by Tony Hunter in consultation with three members of the Board of Reviewing Editors. Although the work is of interest, we are not convinced that the findings presented have the potential significance that we require for publication in eLife.*

We thank the reviewers for their assessment that the work is of interest.

We believe that the findings do have the potential significance required for the publication in eLife and address this concern, in addition to the specific concerns of the reviewing editors, below. We have made the appropriate changes to the manuscript itself, including 12 new experiments (4 new GFP cellular trapping experiment, 2 tethering experiments, 4 ITC measurements and 2 NMR experiments) and 8 revised figures.

*The main linkage between this paper and the original paper in eLife, which justifies this as a Research Advance, is the crystal structure of the Ki-67 motif bound to PPI.*

We appreciate that the reviewers agree that this work is justified as a Research Advance due to its original link with Booth, et al. (Booth et al., 2014).

*This new structure is very similar to that of the homologous RepoMan motif bound to PP1γ, which would have been predicted based on the very strong similarity of the PP1-binding motifs of these proteins.*

As stated above, this statement is incorrect. This is because the RepoMan:PP1 holoenzyme structure has never been published (3 different PDBids for all novel structures are provided in the manuscript). Rather, it is this manuscript reports both the Ki-67:PP1 and RepoMan:PP1 holoenzyme structures—i.e., they are both new structures that were not known prior to this work.

Furthermore, although the ~30 aa sequence conservation between Ki-67 and RepoMan had been identified in the original paper, it was not known until our study that this sequence corresponds to the full PP1 interaction domain for both proteins. The only residues previously identified to interact with PP1 correspond to the well-established RVxF motif. Thus, it is this Research Advance manuscript that demonstrates that residues beyond the RVxF motif are essential for binding PP1 and that they bind PP1 in a manner never observed for any other PP1 regulator. This is a major advance for the field and is why we feel that the finding presented have the significance required for publication in eLife.

*In the rest of the manuscript the authors used the RepoMan:PP1 interaction as a proxy for the Ki-67:PP1 interaction, but without validating any of their conclusions on Ki-67 itself, which makes the link to the original eLife paper rather tenuous.*

We discovered (and describe in this manuscript) that the Ki-67/RepoMan PP1 interaction motifs are exceptionally well-conserved, bind PP1 with identical affinities and bind PP1 via an identical mechanism, as demonstrated by our multiple PP1 holoenzyme crystal structures (Ki-67:PP1γ, RepoMan:PP1γ and RepoMan:PP1α). These data already strongly suggest that the results obtained from RM will be valid for Ki-67. However, we understand the concern of the reviewers and thus have repeated all of the experiments using Ki-67 to experimentally demonstrate this is the case.

1) GFP-trapping experiments. We performed new GFP-trapping experiments with both wt PP1 isoforms (PP1γ and PP1α) and PP1 isoform variants (PP1γR20Q and PP1αQ20R). The results show that only PP1γ and PP1αQ20R effectively trap endogenous Ki-67 (Figure 2). These results mirror that obtained for RM (Figure Supplement 4c).

2) Tethering experiments. We performed new tethering experiments with GFP-Laci-Ki-67 and RFP-PP1 (PP1γ or PP1γR20Q) in Chicken DT40 cells. The results show that only PP1γ co-localizes with GFP-Laci-Ki-67 at the LacO array (Figure 2). These results mirror what was observed for RM (Figure 2).

3) ITC experiments. We performed new ITC experiments with Ki-67 and PP1α, PP1γR20Q and PP1αQ20R. We also performed ITC experiments with Ki-67S507D and PP1γ. The results show that the affinity of Ki-67 for PP1α is weaker than that for PP1γ, the affinities reverse with PP1 variants PP1αQ20R and PP1γR20Q, and that the phosphomimetic mutation of Ki- 67S507D profoundly weakens the affinity of Ki-67 for PP1γ (Table 1, Figure Supplement 1). These results mirror what was observed for RM (Table 1, Figure Supplement 1).

4) NMR/phosphorylation experiments. We performed new NMR and mass spectrometry experiments with Ki-67 and Aurora B kinase. The results demonstrate that only Ki-67 residue S507 is phosphorylated by Aurora B kinase (Figure 3). This result mirrors that obtained for RM (Supplemental Figure 4).

*The Ki-67 and RepoMan motif-PPI structures are quite interesting because the peptide forms a β-hairpin, with its N-terminal part being positioned on the usual RVXF binding site of PP1, but with its C-terminal part turning around 180 degrees, in contrast to other PP1 structures where the polypeptide chain continues more or less straight on the PP1 surface, often establishing a second main contact at the SILK binding site.*

We agree that these structures are “quite interesting”. This is the first time such an interaction has ever been observed with PP1, the phosphatase that dephosphorylates more than 50% of all ser/thr residues in humans. Although PP1 has been studied for decades, a mechanistic understanding of how its regulators bind and direct PP1 activity is only just now beginning to be understood. Prior to our study, the only residues in RepoMan and Ki-67 identified to bind PP1 were the canonical RVxF motif. Here, we used NMR spectroscopy and ITC to identify the full PP1 binding domain, which revealed that the PP1 interaction domain extends beyond the RVxF interaction motif to bind PP1 via a novel mechanism and at a novel interaction site that not only enhances binding, but also mediates PP1 isoform selectivity. We then determined the crystal structures of three different holoenzyme complexes to demonstrate that this is the case: (1) Ki- 67:PP1γ, (2) RepoMan:PP1γ and (3) RepoMan:PP1α.

*However, whether Ki67 and RepoMan are the only ligands for this new binding site on PP1, as the authors claim, is not clear, as it may well be that motifs with divergent or even unrelated sequences bind at the same site.*

Over the last ~15-20 years it has been well established that the PP1 binding pockets (such as the RVxF, SILK and Arg binding pockets) bind to motifs with very limited variation in sequence (Choy et al., 2014). For example, the RVxF motif of known PP1 interactors contains a ‘V’ in 94% of the sequences (the remaining 6% contain an ‘I’) at the ‘V’ position and an ‘F’ in 83% of the sequences (‘W’ in the remaining 17%) at the ‘F’ position. Similarly, the SILK motif varies only in the first residue, being either G or S (i.e., [G/S]ILK). Thus, contrary to the hypothesis put forward above, the protein-protein interaction sites in PP1 have, in fact, strict requirements for the sequence motifs that bind at these sites.

Indeed, this is identical to what is observed for substrates and regulatory proteins that bind calcineurin (which bind PxIxIT and LxVP motifs) (Grigoriu et al., 2013), PP2A (LSPI motifs) (Qian et al., 2015) and also defines the sequence specificity of multiple enzymes, like kinases (e.g. PKA, RRxS/TΦ, CDK, S/TPxK/R, or the D-motif for substrate recognition of MAPKs, among 10’s and 100’s of other examples). This demonstrates that protein-protein interaction pockets bind to specific sequence motifs and, in this way, achieve specificity. Thus, there is no reason not to expect the same strict sequence requirements will be required for binding the KiR-SLIM binding pocket on PP1. Indeed, this is the reason why numerous efforts are ongoing to target these sequences in the pharmaceutical industry (including Vertex Pharmaceuticals & Reflexion Pharmaceuticals) and academia.

*Regulation of RV(S/T)F sites by Aurora B kinase has already been established (most notably, for the N-terminal region of Knl1/CASC5), as has the control by Aurora B of RepoMan recruitment to chromatin.*

The specificity of phosphorylation events in time and space are critical for the exceptional fidelity of phosphorylation signaling (arguably, it is this reason why the 100s of phosphoproteomics studies performed in the last decade have been so informative). Our observation that Aurora B kinase specifically phosphorylates Ki-67/RepoMan at their RVxF motifs is a key novel discovery as it explains how and when these holoenzymes assemble during the cell cycle.

Although some RVxF sites have previously been demonstrated to be regulated by phosphorylation (i.e., PNUTS and Gm by PKA; KNL1 by Aurora B kinase), this manuscript describes the first time that the interaction of RepoMan or Ki-67 with PP1 has ever been demonstrated to be regulated by Aurora B kinase. This is important because it reveals a new mechanism by which the assembly of the Ki-67/RepoMan holoenzymes are regulated during the cell cycle. Namely, the phosphorylation of Ki-67 and RepoMan by Aurora B kinase explains why Ki-67 is maintained in a phosphorylated state in early mitosis and why the RepoMan:PP1 holoenzyme is largely excluded from chromatin prior to anaphase onset–neither can bind and form stable holoenzymes with PP1. As noted by the reviewers, Aurora B augments this effect by phosphorylating a second site on RepoMan (Ser817; ~400 amino acids away from the RVxF 5 motif) to further prevent it from binding chromatin. Thus, our discovery that Aurora B phosphorylates a second site, the RVxF site, is not only novel, it also demonstrates that Aurora B phosphorylation of RepoMan results in two distinct mechanisms for reducing the premature targeting of PP1 to chromosomes prior to anaphase onset. Both of these are important and our results provide a new understanding of how these holoenzymes are regulated during the cell cycle.

*Moreover, no direct evidence of the functional importance of the reported RepoMan and Ki-67 interactions with PP1γ for progression through mitosis is provided, for instance as assessed by expressing appropriate mutants in knockdown cells.*

In fact, just the opposite is true. Namely, our structural, biochemical and cell biological studies now explain a number of pieces of data concerning Ki-67 and RepoMan the functional importance of Ki-67 and RepoMan during the cell cycle.

First, previous work has demonstrated that overexpressing a RAXA mutant of RepoMan (which inhibits PP1 binding; see Figure 3 in the current manuscript) in HeLa cells results in the displacement of PP1γ from chromatin. This results in an adverse effect on cell viability, as almost no mitotic cells and very few interphase cells are detected expressing high levels of this mutant and a much larger level of cell death was observed (Trinkle-Mulcahy et al., 2006). Similarly, a study by one of the authors of this manuscript demonstrated that phosphorylation of RepoMan at three distinct sites (Ser400, Ser412 and Ser419) also inhibits PP1 binding (Qian et al., 2015).

Conversely, mutating these residues to alanine (non-phosphorylatable; 3A variant) results in the binding of PP1 to RepoMan prior to anaphase onset (these mutants were expressed in knockdown cells) and causes the precocious dephosphorylation of mitotic-exit substrates. Our work reveals also how this functional result is achieved at a molecular level--namely, that all three of these residues are part of the PP1 interaction motif and thus, their phosphorylation directly inhibits PP1 binding. Taken together, these data demonstrate the essential role of RepoMan in cell viability is directly related to its PP1-targeting activity.

Second, when endogenous Ki-67 is replaced by a Ki-67 RASA mutant (which cannot bind PP1), the accumulation of PP1γ on chromosomes is reduced (Booth et al., 2014; Takagi et al., 2014). However, the formation of daughter nuclei are not notably affected. This mild phoenotype is consistent with that observed for cells in which Ki-67 is knocked-down completely. Thus, Ki-67 is less important for mitosis and viability than is RepoMan.

Third, both Ki-67 and RepoMan were previously demonstrated to be specific for the gamma isoform of PP1 (Booth et al., 2014; Trinkle-Mulcahy et al., 2006). We show unequivocally that this specificity is due to the recognition of a single residue within the PP1 catalytic domain by these regulators, Arg20. This is a fundamental new discovery that now explains a plethora of other papers that observed but did not explain this isoform specificity e.g. for nucleolar targeting.

In conclusion, the importance of these regulators in cell viability and mitosis, in spite of their conserved PP1 interaction motif, differ. This is because the only residues that the two proteins have in common are the short ~30 amino acid PP1 interaction motif; the rest of the sequences are divergent. As stated above, knock-down studies with RepoMan demonstrate that it is critical for viability. The importance of PP1 binding for this function of RepoMan was demonstrated by transfecting a PP1-deficient mutant (RAxA) which led to a large level of cell death. In contrast, 6 similar studies with a Ki-67 RAxA mutant resulted in mild phenotypes, similar to those observed from cells with Ki-67 knocked down. We have edited the text accordingly to more explicitly address these points.

[Editors’ note: the author responses to the re-review follow.]

*Essential revisions:*

*Reviewer 1:*

*Second paragraph of Introduction section: A recent paper in Nature sheds light on an important function of Ki-67 and the authors might elect to cite it.*

We have included a sentence about the newly discovered function of Ki-67 and have also cited this paper.

“Recently, it was shown that one role of Ki-67 is to function as a ‘DNA surfactant’, preventing individual chromosomes from collapsing into a single chromatin mass upon nuclear envelope disassembly by binding directly to the surface of chromatin^12^. A second recently discovered function is that Ki-67 binds and regulates the activity of PP1 during mitosis using the canonical PP1 RVxF small linear motif (SLiM)^1^.”

*Sixth paragraph of Results section: "[…]pull down PP1γ[…]": are the data shown anywhere? They are not in Figure 3*

The pull-down data is published in reference 27 (Qian et al., 2015, Nature Communication). The sentence was previously written as the following to indicate this fact:

“Mutating these residues to phosphomimetics inhibits PP1γ binding, as evidenced by the inability of EGFP-RepoMan3D to pull-down PP1γ from non-synchronized HEK293 cells and […]”

*The discussion on binding specificity of RM and Ki-67 for isoforms of PP1 is limited to the α and γ subunits, but the authors do not mention the β subunit anywhere. This point should be discussed.*

Ki-67 binds to both PP1γ and PP1β. This is not a surprise, as PP1β has the same primary sequence as PP1γ in the newly identified selectivity sequence – only PP1α is different.

This now explains data presented in the *eLife* article on which this research advance is associated with (Reference 1). Namely, Booth et al. (2014) used a tethering/recruitment assay (such as that illustrated in Figure 2) to show that LacI-Ki-67_301-700_ recruits both PP1γ and PP1β to a LacO array in DT40 chicken cells. This work also showed that PP1γ is recruited more efficiently than PP1β. Furthermore, pull-down experiments coupled with SILAC to identify RepoMan interactors whose binding was enriched in anaphase also identified both PP1γ and PP1β, but not PP1α (see reference 21). Finally, we now also show in this current report that PP1β co-immunoprecipitates with RepoMan and this binding is reduced when the corresponding arginine is mutated to Q (R19Q); this data is now included in “Supplemental Figure 5”..

To clarify this in the text, we added/edited the following sentences:

“in vivo, Ki-67 and RepoMan bind specifically to the β- and γ-isoforms, but not the α-isoform, of PP1 (as PP1γ is recruited more efficiently than PP1β, we focused our study on the PP1α and PP1γ isoforms).

GFP-traps showed that GFP-PP1γ and GFP-PP1α^Q20R^ bind endogenous Ki-67 in prometaphase-arrested cells, while GFP-PP1α and GFP-PP1γ^R20Q^ bind much more weakly (Figure 2. Similar GFP-traps showed that GFP-PP1γ, GFP-PP1β and GFP-PP1α^Q20R^ bind both endogenous and ectopically expressed RepoMan in prometaphase-arrested cells, while GFP-PP1α, GFP-PP1γ^R20Q^ and GFP-PP1β^R19Q^ bind much more weakly (Figure 2—figure supplement 3).”

*Fifth paragraph of Results section: "GFP-Traps showed that GFP- PP1γ and GFP-PP1αQ20R bind […] in prometaphase-arrested cells". This is an important point. If the authors' theory on phosphoregulation of the interaction is correct, there ought to be no binding in prometaphase. Please clarify.*

The binding of PP1 to RepoMan is indeed much less prominent in (pro)metaphase than in anaphase because the binding of PP1 is then opposed by phosphorylation of the PP1-binding domain of RepoMan by Cdk1 (see reference 29) and Aurora B (this work). However, the PP1-RepoMan interaction in (pro)metaphase, although less pronounced, can clearly be demonstrated and is essential to dephosphorylate histone H3T3 at the chromosome arms and thereby contributes to the centromeric targeting of Aurora B (see reference 29) and also “Figure Supplement 4”.

*Is Panel 3C cited anywhere in the text?*

Panel 3C was cited in the text (subsection: “Dynamic regulation of PP1y recruitment by Aurora K Kinase”), but an adjacent supplemental citation was referenced incorrectly making it hard to read. This has been fixed.

“Using in vitro phosphorylation assays coupled with NMR spectroscopy and mass spectrometry, we showed that both S507_Ki-67_ and T394_RM_ (Figure 3; Figure 3—figure supplement 2) are phosphorylated directly by Aurora B kinase.”

*First paragraph of Conclusion section: There is additional prior art on the role of PP1 in the control of mitotic checkpoint function, in particular the paper from the Funabiki laboratory (Rosenberg et al. Current Biology 21:942). Please cite it.*

This paper, along with a review from the same laboratory, is now referenced in the introduction in the first paragraph:

“However, how these events are coordinated during mitotic exit is still an open question. The emerging picture is that exiting mitosis requires the specific engagement and activation of protein phosphatases, including ser/thr phosphatase protein phosphatase 1 (PP1)^3,4^.”

*Panel 2F: The upper left panel is probably not the WT Ki-67 sequence, but the positive control GFP-LacI. In panel G, the authors could clearly indicate that they are comparing GFP-LacI to GFP-LacI-Ki67.*

*Please add molecular weight markers to all gels.*

As requested, MW markers have been added to the gels.

*Reviewer 3:*

*1) The statement beginning in the first paragraph of the Results section. ' […] this conserved region[…] constitutes the full PP1 interaction domain'. I could not find evidence for this statement.*

The evidence for this statement is that extending this domain did not enhance binding, as determined by ITC (i.e., RM_383-441_, RM_348-450_ and RM_303-515_ had K_D_s of 117 nM, 124 nM and 77 nM, respectively). These measurements are listed in Table 1. The sentence in which they are referred to reads as follows:

“We also showed that extending this domain does not enhance binding (Table 1).”

This data was further corroborated by NMR spectroscopy as also described in the manuscript.

*2) Also in the same sentence, given the data presented at this point, the comment that 'Ki-67 and RM interact with PP1 using identical mechanisms' is unsubstantiated. Their structural data presented later does give credence to this view.*

We appreciate the reviewer’s insights and agree that in spite of the similar sequences and binding affinities, without the structural data the view that both Ki-67 and RepoMan interact with PP1 using identical mechanisms is premature. Hence, we have edited this sentence to replace ‘demonstrate’ with ‘suggest’.

“Together, these data suggest that Ki-67_496-536_ and RepoMan_383-423_ bind PP1 using identical mechanisms and that this conserved region, which extends beyond the RVxF SLiM, constitutes the full PP1 interaction domain.”

*3) Figure 1 should indicate the Kd and the name of the interaction proteins.*

Edited as requested.

*4) Figure 1 should also indicate the Kd and the name of the interaction proteins and also indicate what ΔKiR-SLiM actually is.*

Edited both the figure and the legend as requested.

*5) Third paragraph of Results. The authors identify a new SLiM using HMMER. Did the authors test the role of consensus residues in this motif, for example F517/F404, P523/A416?*

We tested the role of RepoMan variant F404A in vivo (as measured by monitoring H3T3 phosphorylation). The data show that, alone, this mutation is not sufficient to negatively impact RepoMan function. Thus, the binding strength provided by the additional portions of both the KiR-SLiM and the RVxF/ΦΦ SLiMs are sufficient to overcome this mutation in vivo. However, as we show in Figure 1, removing the SLiM altogether does negatively impact binding, demonstrating its importance.

*6) The authors suggest that Arg20 of PP1γ confers selectivity for RM and Ki-67. The equivalent residue in Gln in PP1α. Unfortunately neither the structure shown in Figure 2 nor the text explains how Arg20 confers this selectivity. What residues on RM and Ki-67 does Arg20 interact with? Are these conserved between RM and Ki-67?*

The mechanism is as follows. The Arg20 sidechain forms: (1) a salt bridge with PP1 residue Glu77 and (2) makes a strong planar stacking interaction (cation/π interaction) with PP1 residue Phe81. Important – neither interaction is possible with the uncharged, shorter Gln20 residue in PP1α. Thus, Arg20 stabilizes the interaction between PP1 loop 1 (residues 19-26) and the remainder of the PP1 catalytic core. This is consistent with our previous data – in the multiple PP1α structures we have determined to date (4XPN, 4MOV, 4MOY, 4MP0, 3V4Y, 3EGG, 3EGH, 3HVQ, 3E7A, 3E7B), the density for PP1α loop 1 is generally the weakest in the entire structure (even in the 1.4 Å maps, 4MOV). In contrast, in PP1γ structures, the electron density of this loop is consistently better. This is also consistent with the B-factors of this loop: the average increase of the main chain B-factors for this loop (normalized against the average B-factor) are consistently higher for PP1α than PP1γ. Together, these data demonstrate that only in PP1γ is this defined pocket is readily available for binding, which, as a consequence, leads to the isoform specific interaction of RM and Ki-67 for PP1γ.

This is now more clearly described in the text (Results section, subsection: “The specific recruitment of PP1y by Ki-67 and RepoMan”).

“Although R20 does not interact with RepoMan directly, it confers selectively through it interactions with PP1 which order the L1 loop. Namely, Arg20 sidechain forms a salt bridge with PP1 residue Glu77 and makes a planar stacking interaction (cation/π interaction) with Phe81. Neither interactions are possible with Q20, as the side chain is both uncharged and too short. Thus, only in PP1γ is this pocket ordered and readily available for binding, which allows for the isoform specific interaction of RM and Ki-67.”

*7) The labels in Figure 3 aren't clear.*

The tops of the labels of Figure 3 were inadvertently cut-off and have been fixed.